# An evaluation tool to strengthen the collaborative process of the public-private partnership in the veterinary domain

Mariline Poupaud[1,2]*, Nicolas Antoine-Moussiaux[2], Isabelle Dieuzy-Labaye[3], Marisa Peyre[1]

**1** UMR ASTRE, Univ Montpellier, CIRAD, INRAE, Montpellier, France, **2** Fundamental and Applied Research for Animals and Health (FARAH), University of Liège, Liège, Belgium, **3** World Organisation for Animal Health (OIE), Paris, France

* mariline.poupaud@cirad.fr, mariline.poupaud@gmail.com

**Data Availability Statement:** If readers would like to access to the spreadsheet file of the case-study in Ethiopia, the results of the expert elicitation, or the transcripts of the scoring process with the

## Abstract

Public-private partnerships (PPPs) in the veterinary domain are widely implemented world-wide and can help to strengthen the capacities of Veterinary Services. Few analyses have been made of these initiatives. This study is aimed at developing an evaluation tool based on participatory approaches and focusing on the quality of PPP processes in the veterinary domain. The tool was divided into ten sections relevant to PPP process organisation and activities. The 44 evaluation criteria and six quality attributes (operationality, relevance, acceptability, inclusiveness, adaptability, and stability) were identified based on literature review and case-study application. The tool was adjusted during four regional PPP training workshops bringing together stakeholders from both public and private sectors. Finally, the tool was validated through an experts' elicitation process and applied in the field in Paraguay. The tool was developed in a non-normative perspective to help the partners adapt the PPP to their specific context, to maximize the opportunities and minimize the risks of such collaborations, and to formulate adapted recommendations to strengthen and improve the PPP collaborative process and thus the outcomes. In an ex-ante perspective, this tool would also help public and private actors to engage and develop a PPP process following the best possible practices. The aim of this tool is to help decision making in terms of PPP development and implementation in the veterinary domain to ensure the added value and relevance of such a collaborative approach in different countries worldwide.

## Introduction

Public-Private Partnerships (PPPs) in the veterinary domain (as defined in the *Terrestrial Animal Health Code* [1]) are "a joint approach in which the public and private sectors agree on responsibilities and share resources and risks to achieve common objectives that deliver benefits in a sustainable manner" [2]. The Performance of Veterinary Services (PVS) Pathway, a flagship program proposed by the World Organisation for Animal Health (OIE) for evaluating

different partners in Paraguay, the data are available upon request from the authors: mariline. poupaud@cirad.fr. The two questionnaires of the expert elicitation are available in the Supporting information.

**Funding:** This study was carried out as part of a doctoral thesis funded by the French Agricultural Research Centre for International Development (CIRAD) and the OIE Public Private Progress project which is supported by the Bill & Melinda Gates Foundation under the grant number: OPP1159705.

**Competing interests:** The authors have declared that no competing interests exist.

and advising on policies and strategies to strengthen national Veterinary Services (as defined in the *Terrestrial Animal Health Code* [3]), recognises PPPs as a potential tool for such strengthening [4].

From the analysis of 97 initiatives implemented across the world, Galière et al. (2019) [5] highlighted that PPPs in the veterinary domain involve a diversity of actors, mechanisms and objectives and can be grouped into 3 main clusters [5]. Cluster 1, "transactional PPP" are often initiated and financed by the public sector and the services come from private veterinarians or paraprofessionals who are contracted or given a sanitary mandate. Cluster 2, "collaborative PPP", corresponds to PPPs usually motivated by trade, exports and/or commercial interests, initiated by both the private sector, often represented by producer associations, and the public sector. Finally, Cluster 3 "transformative PPP" corresponds to PPPs focused on establishing capability and development objectives, initiated and financed by the private sector (local or international companies). Ahuja (2004) [6], analysing the economic rationale of sector roles in the provision of animal health services, stressed the importance of a division of labour between the public and private sectors. For example, with regard to animal health services in remote areas, it encourages working through civil society organisations, and using para-professionals and community-based animal health service delivery systems [6].

Despite many examples of PPPs implemented in the field in the veterinary domain, few studies have evaluated the initiatives in place [7]. Evaluation is a means to reinforce partnerships and the process of collaboration. It helps in planning, redefining strategies, taking appropriate corrective actions, ensuring trust between partners, optimizing resources and finally ensuring the effectiveness of actions [8, 9].

However, no evaluation framework of PPPs in the veterinary domain has been formulated [7]. The evaluation frameworks in Public Health highlight the importance of evaluating the PPP process and not only its outcomes, by analysing the quality of the mechanism and functioning of PPP. Analysis of these evaluation frameworks has identified the important steps in evaluating the PPP process: analysing the PPP objective(s), the governance process, the planning process and the collaboration process between partners [7]. For example, they emphasized the need for partners to understand their respective motivations and objectives [10]. The quality of PPP outcomes will depend on the quality of its organization. Hence, the evaluation of the PPP process is crucial to providing recommendations on how to improve the PPP's outcomes. Evaluation of animal health programs does not usually include an analysis of the process. To our knowledge, the only two existing tools focusing on the process are specific to surveillance programs. The Oasis tool assesses the functional parts of a surveillance system [11] and the One Health matrix assesses the multi-sectorial collaboration in One Health surveillance programs [12]. The Oasis tool model has been used to evaluate many surveillance systems and has demonstrated its ease of use.

The PPP process evaluation frameworks in Public Health provide a robust basis, but need to be adapted to the veterinary domain by including specific key success factors and obstacles identified in PPPs in this domain, and could be expanded towards a more integrated approach.

PPPs represent a means to achieve objectives and can be transitional; they need to be adapted to their own context and they cannot be reduced to "a formula" to be applied and followed [10]. This is why we argue that PPP evaluation should mobilize an evaluative research approach that seeks to understand the how and why of the results, rather than a normative evaluation approach that seeks to compare the components of the intervention to standards [13]. There is general agreement in the literature that PPPs need to present collaborative advantages; that is, they should represent an added value compared to a program that does not involve PPPs [7]. However, it is not easy to measure the benefits of collaboration. It is

recognised that the best way to do so is to engage in deliberation among partners about this potential added value, using participatory approaches [14]. Furthermore, participatory approaches to evaluation have proven very useful in ensuring the adaptability and acceptability of the evaluation outputs, facilitating the implementation of corrective actions to improve process quality [15, 16]. To the best of our knowledge, no tool has yet been developed to allow a participatory evaluation of the quality of the PPP process in the veterinary domain.

The aim of this study is to create a participatory tool that focuses on the PPP process in the veterinary domain. The intended tool would help in formulating recommendations to strengthen the collaborative process and thus improve the outcomes. In an ex-ante perspective, this tool would also help to anticipate a collaborative process.

## Material and methods

### Tool organisation and development

The tool was developed on the basis of existing tools—such as the Oasis tool which aims to evaluate the quality of the animal health surveillance system process [11] and Survtool [17] which assesses the strengths of collaborations within One Health surveillance systems. The tool is comprised of sections, representing PPP process organisation and activities. Each PPP process section is assessed using a set of evaluation criteria, each evaluation criterion being scored on a four grade scale from 0 to 3. The influence of the PPP process on its performance is assessed using quality attributes.

The PPP process sections, evaluation criteria to assess each PPP process section and the quality attributes which represent overall PPP performance were defined according to the literature review and PPP case-study analysis. The first version of the tool was tested during 4 regional PPP training workshops organised by the OIE in Africa and Asia, and the tool was amended based on user feedback. The revised version (version 2) of the tool was validated through an experts' elicitation process (Fig 1).

In parallel, a checklist was created to support the collection of useful information to be used for the scoring of the evaluation criteria, together with a scoring guide to help the evaluators correctly understand the evaluation criteria and facilitate the scoring process. Finally, a spreadsheet was developed to integrate the evaluation criteria scores and automatically process calculation of the PPP process sections and quality attributes [11].

### Literature review and case study analysis to define the sections of the public-private partnership process, evaluation criteria and quality attributes

The sections of the PPP process and evaluation criteria, identified in a scoping review that analysed the existing evaluation frameworks of PPPs in the veterinary domain and public health, were used to construct the first version of the tool [7]. In addition, the OIE PPP Handbook of best practices, co-constructed with actors involved in PPPs or catalysers of PPPs (individuals or organisations whose activities support or enable the implementation of PPPs), was used to identify the PPP process sections of the tool [2]. Evaluation criteria used in the Oasis tool and One Health matrix to evaluate the process of surveillance programs were also analysed to identify additional evaluation criteria to include in the PPP tool [11, 12]. Indeed, as for PPP, surveillance systems are a collaboration of multiple actors from different sectors and with different perspectives.

Finally, in order to select the quality attributes of the PPP performance, the attributes from the One Health matrix were compared to the theoretical framework developed by Bryson and

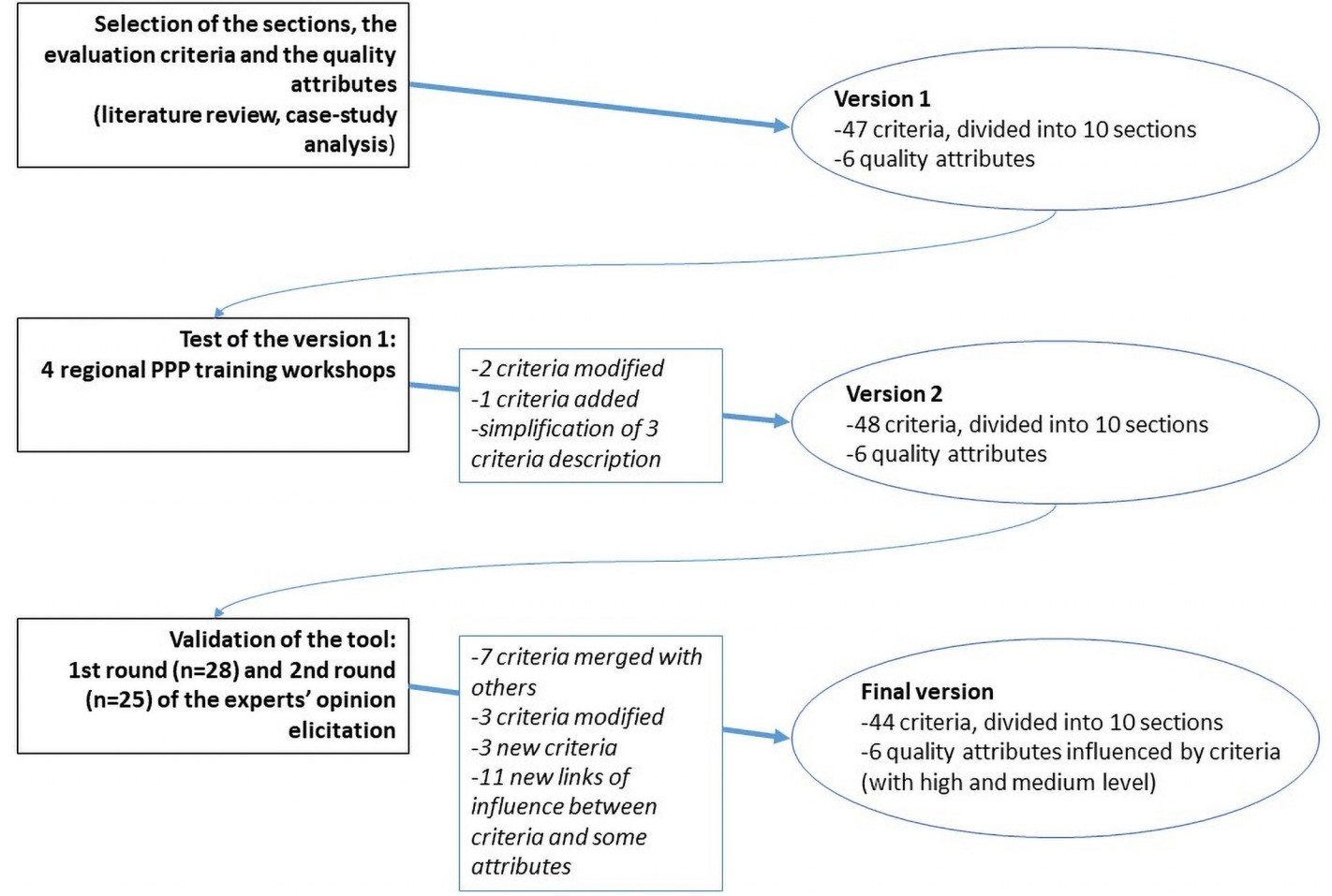

**Fig 1. The process of the tool development.** The different steps of this study captured the viewpoints of public and private partners, catalyzers and actors impacted by the public-private partnerships. PPP: public-private partnership.

collaborators (2015) [14] on cross-sectoral collaboration that includes public-private partnerships in the Public Affair domain.

The evaluation criteria were also defined using the results of a PPP evaluation case study performed within the framework of the OIE PPP initiative [18]. This case study addressed a long-term public-private partnership in the veterinary domain, between a poultry producing company and the Ethiopian Ministry of Livestock and Fisheries, aiming at developing the poultry sector in Ethiopia [19]. This evaluation case study was conducted with the participation of the different categories of actors involved, i.e. public and private actors from national and local levels. Semi-structured individual interviews (n = 33) addressed the topics of the context of implementation, organisation and process of the PPP, the strengths and weaknesses of the system, the actors involved, the missing actors, and the prospects for improvement. In addition, two participatory workshops were held with the different stakeholders to validate the results obtained, compare the different viewpoints of stakeholders and co-develop improvement scenarii (n = 26 and 53). Every discussion that took place during the workshops or individual semi-structured interviews was recorded and transcribed. The transcripts were read, and categories emerged from the reading, corresponding to the functional process of the PPP (such as type of private partner, type of public partners, training organization etc). During a

second reading of the transcripts, the qualitative data were classified into these categories in a spreadsheet file. Actors' narratives were used to identify which evaluation criteria selected from the literature were applicable to this case study and which evaluation criteria were missing.

## Public-private partnership regional training workshops to test version 1 of the tool

Version 1 of the tool was tested and improved during four regional training workshops on PPPs organized by OIE. One workshop was held in Ethiopia for English-speaking African countries, another in Tunisia for French-speaking African countries, another in Nepal for South Asian countries and the last one in Thailand for South-East Asian countries. The four workshops involved around 200 public and private stakeholders who were engaged in PPPs or who were planning to set up a PPP initiative. Participants were from national Veterinary Services, producer associations, private veterinary workforce associations, private industry (meat, dairy or veterinary products) and non-governmental organizations. The tool was tested by groups of 5 to 10 people, mixed between public and private sectors, during a one-hour session. Participants were asked while implementing the tool to review the relevance of the evaluation criteria used, the clarity of evaluation criteria description, to identify any missing evaluation criteria, to comment on the usefulness of the tool, and how easy it was to use. Participants' feedback was collected and analysed to produce Version 2 of the tool and a revised list of evaluation criteria and associated definitions.

## Experts' elicitation process to validate the tool (version 2)

The tool was validated by experts' elicitation in a two-round process, consisting of two online-questionnaires developed with the Surveymonkey® tool that experts have to fill in. The aims of this experts' elicitation process were: 1) to validate the evaluation criteria (relevance, definition, exhaustiveness) used to assess the strengths and weaknesses of each section of the PPP process and 2) to validate the influence of each criterion on quality attributes of the PPP performance. The first questionnaire was sent on the 15th of September 2020 to 37 experts, and closed on the 1st of October; 27 experts responded to it with a mean time of 43 minutes (from 21 min to 2 hours and 25min). The 27 experts were private partners (e.g. private companies, private veterinarians or veterinary associations, producer organizations) (n = 8), public partners from the official Veterinary services (n = 3) and catalysers from international organizations such as OIE, Food and Agriculture Organization of the United Nations (FAO), and the International Fund for Agricultural Development (IFAD) (n = 16). The experts had been involved in PPPs or supporting PPPs for less than 2 years (n = 6), from 2 to 5 years (n = 7), from 5 to 10 years (n = 9), or for more than 10 years (n = 6).

The results were analysed and any discrepancies between experts were reviewed during a second round. The second questionnaire was sent to the same 27 experts on the 28th of October and closed on the 13th of November; 25 experts (two experts from the catalysers did not answer during the second round) responded with a mean time of 24 minutes.

The questionnaires from the two rounds included four main parts: (i) background information on the experts, (ii) review of the PPP process sections and evaluation criteria, (iii) review of the quality attributes, and (iv) review of the influence of the evaluation criteria on the quality attributes (S1 and S2 Files). The two questionnaires were tested through one pilot interview each. In parts 2 and 3, the experts were asked to review the relevance of the evaluation criteria (yes/no) and if they could identify missing ones. In part 4, the experts had to review the level of influence (no influence/ low level/ medium level/high level of influence) of the evaluation

**Table 1. Calculation of the weighted percentage of experts used for the analysis of the experts' elicitation.** The weights represent the level of confidence of the experts in their answers. This calculation was used to validate the level of influence of evaluation criteria on the six quality attributes.

| Level of influence of an evaluation criterion on a quality attributes | Percentage of experts | Weight = level of confidence (from 0 to 1) | Weighted percentage of experts |
|---|---|---|---|
| High | a1 | w1 | $= a1^*w1/ \sum ai^*wi$ |
| Medium | a2 | w2 | $= a2^*w2/ \sum ai^*wi$ |
| Low | a3 | w3 | $= a3^*w3/ \sum ai^*wi$ |
| No influence | a4 | w4 | $= a4^*w4/ \sum ai^*wi$ |

criteria on the quality attributes and to provide the level of confidence in their answers (0 = not confident; 0.5 = quite confident; 1 = very confident). Experts' answers were then uploaded into an spreadsheet, a descriptive quantitative analysis was conducted for each answer, open comments and justifications about their selection of evaluation criteria and attributes were analysed.

The evaluation criteria and the quality attributes were validated if 85% or more of the experts considered them to be relevant. Experts' comments were used to improve or clarify the evaluation criteria definitions. The evaluation criteria not validated according to this threshold, were revised based on experts' comments and included in the second round. The percentage of experts who selected each levels of influence of the evaluation criteria on the quality attributes (high, medium, low, no influence) were weighted according to the level of confidence of the expert in their answers (Table 1).

The level of influence was validated when the agreement between experts reached more than 50% of the weighted percentage. If not, they were included in the second round. If no agreement was reached after the second round, the intermediary level of influence was selected.

## Field testing of the tool in Paraguay

The tool was implemented in a PPP in Paraguay for the control of foot-and-mouth disease (FMD). The tool was implemented through an external actor who is part of the research team, with groups of 3 to 7 people who were public and private partners. This was done at national level (n = 3) with actors in charge of the national program and at local level, in two different localities (n = 5 and n = 7), with actors in charge of the program implementation in their localities. For each evaluation criterion, the actors had to agree on a grade. If they did not agree, they were asked to explain why they selected such a grade. They were then asked to find a consensus (e.g., a score of 1 if some had initially put 0 and the other 2). Each discussion lasted between 1 and 2 hours and was recorded and transcribed.

## Ethics statement

This study does not concern human health and medical research or animal research, hence, no ethics committee was consulted for study approval.

For the case-study in Ethiopia, the approval to implement the participatory study was obtained from the managing director of the private poultry producing company and, the delegate of the OIE in Ethiopia, who is also the Chief Veterinary Officer at the Ministry of Agriculture. The semi-structured interviews and the workshops were carried out after presenting the study objectives and obtaining verbal informed consent from all volunteer participants. The results obtained from this evaluation case-study were presented and validated by the volunteer participants of the second workshop.

For the PPP regional training workshops, the workshops in the four regions were organised in collaboration with the respective regional representation of the OIE (of Africa for the workshops organized in Tunisia and Ethiopia, and of Asia and the Pacific for the workshops organized in Thailand and Nepal), and a permission was asked from each OIE Delegate, often also the Chief Veterinary Officer, of the involved country. In each workshop, when implementing the first version of the tool, explanation were given on the goal of this exercise to the volunteer participants.

For the experts elicitation, a first email was sent to 45 pre-selected experts (from the private sector, the public sector and catalyser groups), based on personal contacts of CIRAD and OIE, mentioning the goal of the study and asking if they were interested in participating. The first questionnaires was sent only to those who mentioned their interest (n = 37), and the second questionnaire only to those who answered to the first questionnaire (n = 27). Feedback from the analysis of the answers given to the two questionnaires was sent to all 27 experts.

For the field testing in Paraguay, the approval to implement the participatory evaluation of the PPP was obtained from the regional representative of the OIE of the Americas, the Deleguate of the OIE of Paraguay, the Chief Officer of the Veterinary Services in Paraguay and the director of the private foundation of the bovine producers. The implementation of the evaluation tool was carried out after presenting the study objectives and obtaining verbal informed consent from all volunteer participants.

No personal information about volunteer participants was requested in any of the studies (Ethiopian case-study, PPP regional training workshops, experts' elicitation and the field testing in Paraguay), the privacy rights of participants were fully protected, and all data were anonymized. Any of the studies included minors.

## Results

### Public-private partnership process evaluation tool organisation

The final version of the tool is composed of 10 sections of the PPP process, representing the organisational process of a PPP and its activities, 44 evaluation criteria and 6 quality attributes, assessing the influence of the public-private partnership process on its performance (Tables 2 and 3).

The scoring guide is presented in S3 File. Four grades were defined for each evaluation criterion: grade 3 indicates that partners are fully satisfied with the criteria, while grade 0 indicates a total absence of satisfaction and 'not applicable' indicates that this criterion is not relevant to the PPP considered. Like the Oasis tool, the spreadsheet comprises three sheets. The grade of the 44 evaluation criteria, once selected, should be captured in the first spreadsheet. The second sheet displays the graphic output 1, a set of pie charts which represent the result of the scores obtained by all the evaluation criteria for each of the corresponding PPP process sections (Figs 2 and 3). Graphic output 1 is considered as a general view of the structure of the PPP process, helping to identify its strengths and weaknesses easily. The third sheet presents the graphic output 2, a spider chart which is the assessment of the six quality attributes. Graphic output 2 represents the influence of the process on the quality of the PPP performance. The result of each quality attribute is the result of the combination of the score of each corresponding evaluation criterion (Figs 2 and 3).

### Selection of public-private partnership process sections, evaluation criteria and quality attributes

Ten PPP process sections, and 47 evaluation criteria were retrieved from the literature analysis.

**Table 2. Presentation of the tool validated by the experts' elicitation: 10 sections of the public-private partnership process, 44 evaluation criteria, and 6 quality attributes.** The sections represent the public-private partnership process organization and activities. Each sections is composed of a set of evaluation criteria. The six quality attributes assess the influence of the public-private partnership process on its performance. The evaluation criteria and the quality attributes were validated if 85% or more of the experts considered them to be relevant.

| PPP process sections | Evaluation criteria | Influence on the quality attributes |
|---|---|---|
| **Section 1: Objective(s) of the PPP** | 1.1 Common objective(s) | **Operationality** |
| | 1.2 Formalization of the common objective | **Stability** |
| | 1.3 Position of the partners regarding this common objective | **Acceptability** |
| | 1.4 Added value of the PPP | **Stability, Relevance** |
| **Section 2: Specific interest and benefits** | 2.1 The specific interest of the different partners | **Relevance, Acceptability** |
| | 2.2 Allocation of benefits and other outputs (ownership) | **Relevance, Acceptability, Inclusiveness** |
| | 2.3 Achievement of goal(s) of the Veterinary Services | **Relevance** |
| | 2.4 Achievement of goal(s) of the private sector | **Relevance** |
| **Section 3: Risks and constraints** | 3.1 Risks and constraints of getting involved in the PPP | **Stability, Adaptability** |
| | 3.2 Allocation of the constraints | **Acceptability, Inclusiveness** |
| | 3.3 Change of practices | **Operationality, Adaptability** |
| | 3.4 Negative cost to the society | **Stability, Relevance** |
| | 3.5 Conflicts of interest | **Stability, Acceptability** |
| **Section 4: Analysis of the context and external factors** | 4.1 Relevance of common objective and of the strategy regarding the context | **Relevance** |
| | 4.2 International, regional, national, and local laws | **Operationality** |
| | 4.3 Potential threats of the PPP and mitigation | **Stability, Operationality** |
| | 4.4 Organisation of private and public sectors | **Stability, Operationality** |
| | 4.5 Analyses of pre-existing PPPs | **Relevance** |
| **Section 5: Governance of the PPP** | 5.1 Formalization of the PPP | **Stability, Acceptability** |
| | 5.2 Knowledge of the terms of the partnership (contract) and endorsement by all the partners | **Stability, Acceptability** |
| | 5.3 Shared decision making process | **Acceptability, Adaptability, Inclusiveness** |
| | 5.4 Opportunities of private parties' involvement | **Adaptability, Inclusiveness** |
| | 5.5 Funding and human resource availability | **Stability, Operationality** |
| | 5.6 Funding and human resource allocation | **Acceptability** |
| | 5.7 Adequacy with the Veterinary Services mandate | **Relevance** |
| **Section 6: Planning and responsibilities of the PPP** | 6.1 Division of roles and responsibilities | **Operationality, Acceptability** |
| | 6.2 Potential other partners | **Stability, Adaptability, Inclusiveness** |
| | 6.3 Inclusion of vulnerable group | **Adaptability, Inclusiveness** |
| | 6.4 Defined duration | **Stability, Operationality** |
| | 6.5 Modalities of implementation of the PPP activities | **Stability, Adaptability** |
| | 6.6 Joint work plan | **Operationality, Adaptability** |
| **Section 7: Competencies and trainings** | 7.1 Confidence in other partners' competencies and satisfaction of partners about their own competencies | **Acceptability, Inclusiveness** |
| | 7.2 Organisation of training and reinforcement of capacities | **Operationality, Relevance, Adaptability** |
| | 7.3 Accessibility and frequencies of trainings | **Operationality, Inclusiveness** |
| **Section 8: Communication and transparency of the PPP** | 8.1 Internal communication | **Operationality, Acceptability, Adaptability, Inclusiveness** |
| | 8.2 Agreement in resolution modalities in case of conflict | **Stability** |
| | 8.3 Communication with other parties, politics, and with end users | **Acceptability, Adaptability, Inclusiveness** |
| | 8.4 Transparency | **Stability, Inclusiveness** |
| **Section 9: Collaboration in the PPP** | 9.1 Willingness to collaborate and partners' acceptance of their own roles | **Acceptability, Inclusiveness** |
| | 9.2 Level of involvement of partners/mobilisation | **Acceptability** |
| | 9.3 Willingness for capacity building in PPPs (existence of a champion?) | **Operationality, Adaptability** |

(*Continued*)

**Table 2.** (Continued)

| PPP process sections | Evaluation criteria | Influence on the quality attributes |
|---|---|---|
| **Section 10: Monitoring and evaluation of the PPP** | 10.1 Internal monitoring of the PPP | **Operationality, Stability, Adaptability** |
| | 10.2 Agreed indicators for joint internal monitoring | **Acceptability, Adaptability** |
| | 10.3 External evaluation | **Operationality, Acceptability, Adaptability** |

PPP: public-private partnership

Two additional evaluation criteria were identified from the Ethiopian case study. The Veterinary Services in Ethiopia have limited numbers of veterinarians specialized in poultry production and many farmers reported having limited knowledge about poultry management.

- "We have general veterinarians; we don't have poultry veterinarians who have good background in poultry. We have few but it's not enough." (interview, poultry production director of the public veterinary service)

This lack of capacity may limit the involvement of some actors in this PPP on poultry production, and an evaluation criterion "confidence in other partners' competencies and satisfaction of partners about their own competencies" was added.

The smallholder farmers mentioned their fear of losing their local poultry breed, explaining why some of them are reluctant to get involved in this program involving an improved chicken breed.

- "There is no consideration in preserving the local genotypes" (interview, Ethiopian farmer)

- "[. . .] smallholders have preference for the local breeds based on their culture. They are used for adoration of ancestors, or for ceremony to solve disputes. [. . .]" (interview, social scientist in International Livestock Research Institute in Ethiopia)

Also, the private poultry producers not involved in the PPP were afraid of losing the production market. They do not allow the private actors of the PPP to access the poultry association.

An evaluation criterion "negative cost to the society" was added. This case study confirmed that it is important to consider all the potential results of the PPP, including the negative ones, which can weaken the initiative.

Six quality attributes (operationality, relevance, acceptability, inclusiveness, adaptability, and stability) were selected based on the functional attributes used in the One Health matrix [12]. Although those attributes are applied to a multi-sectoral surveillance system, they focus on a collaborative process and it appeared appropriate to employ the same vocabulary for the PPP process tool. However, not all of them were appropriate; and to select the most relevant attributes for the PPP process, we compared them to the Bryson framework on cross-sector collaboration that includes public-private partnerships in the Public Affairs domain. This framework emphasizes that "collaborating parties should design processes, structures, and interactions in such a way that desired outcomes will be achieved", which is implied by the **operationality** quality attribute. This framework emphasizes that partners must be sure that "there is a clear collaborative advantage to be gained by collaborating", which is tackled in the quality attribute **relevance**. This framework recommends "use inclusive processes to develop inclusive structures", which relates to the quality attribute **inclusiveness**. Finally, this framework stresses the need to "view collaborations as complex, dynamic, multilevel systems" and to

**Table 3. The six quality attributes of the public-private partnership process and their definition.** Those six quality attributes assess the influence of the public-private partnership process on its performance, and are influenced by different evaluation criteria. The high (score of 10) and medium (score of 5) level of influence of the evaluation criteria on the six attributes were validated during the experts' elicitation as the agreement between experts reached more than 50% of the weighted percentage of experts (see Table 1). The levels of influence that did not reach 50% of the weighted percentage of experts consensus were between medium and high level and a score of 7,5 was given.

| The six quality attributes and their definition | Evaluation criteria with a high level (10) of influence | Evaluation criteria with a level of influence between medium and high (7,5) | Evaluation criteria with a medium level (5) of influence |
|---|---|---|---|
| **Operationality** (influenced by 16 evaluation criteria) The quality attribute of operationality includes the technical aspects of the program (governance, trainings, implementation of activities) and resource management. The governance of PPP is operational, and collaboration is effectively implemented to meet the main objective. Trainings are organised to be sure that stakeholders can fit their roles. The mechanisms for resource allocation are defined. The resources are appropriate and available for the effective implementation of activities. | 1.1, 4.2, 4.3, 4.4, 4.5, 5.5, 6.1, 6.6 7.3, 8.1, 9.3, 10.1, 10.3 (n = 13) | 6.4 (n = 1) | 3.3, 7.2 (n = 2) |
| **Relevance** (influenced by 9 evaluation criteria) PPP strategy, modalities and activities are relevant regarding the main objective. The main objective is relevant and useful regarding the context (epidemiological, institutional, environmental, societal). The PPP represents a clear added value to achieve the objective. | 1.4, 2.1, 2.2, 2.3, 2.4, 4.1, 5.7 (n = 7) | 3.4 (n = 1) | 7.2 (n = 1) |
| **Acceptability** (influenced by 17 evaluation criteria) All relevant stakeholders demonstrate trust in the system, mutual understanding and willingness to collaborate. The objectives and outputs of the PPP meet the stakeholder's expectations. Actors are satisfied with the distribution of resources. The PPPs have societal legitimacy. | 1.3, 2.1, 2.2, 3.2, 3.5, 5.1, 5.2, 5.3, 5.6, 7.1, 8.1, 8.3, 9.1, 9.2, 10.2, 10.3 (n = 16) | (n = 0) | 6.1 (n = 1) |
| **Inclusiveness** (influenced by 13 evaluation criteria) Relevant actors participate in governance mechanisms. Roles in PPP are adequately allocated to actors with regard to their mandates and competencies. At the relevant level, corresponding actors and data sources are considered to meet the collaborative objective(s). PPP provide a trustworthy environment where stakeholders can freely express their views and be heard, creating mutual understanding. The vulnerable group are take into consideration. | 2.2, 3.2, 3.4, 5.3, 5.4, 6.3, 7.1, 7.3, 8.1, 8.3, 8.4, 9.1. (n = 12) | (n = 0) | 6.2 (n = 1) |
| **Adaptability** (influenced by 15 evaluation criteria) PPP can adapt and evolve upon changes in governance modalities, knowledge and context in order to best suit the changing environment. PPP should be flexible to resist over time. PPP activities should be flexible to meet the partners' expectations. The decision-making process should allow for changes within the PPP to enable improvement of the process if deemed necessary. | 5.3, 5.4, 6.5, 6.6, 7.2, 8.1, 8.3, 9.3, 10.1, 10.2, 10.3 (n = 11) | 3.1, 3.3, 6.3 (n = 3) | 6.2 (n = 1) |
| **Stability** (influenced by 16 evaluation criteria) PPP is stable in the time defined by the stakeholders. This means that the PPP is strong enough to withstand external threats, such as changing environment, and continue to operate during the defined duration. The formalisation and endorsement of the agreement satisfied all relevant stakeholders. | 1.2, 1.4, 3.1, 3.4, 3.5, 4.3, 4.4, 5.1, 5.2, 5.5, 6.5, 8.2, 8.4, 10.1 (n = 14) | (n = 0) | 6.2, 6.4 (n = 2) |

PPP: public-private partnership

"adopt flexible governance structures", in line with the **adaptability** quality attribute. The need for adaptability is also acknowledged for PPPs in health system strengthening [10]. Two other attributes presented in the One Health matrix were also selected. **Stability** represents the evaluation criteria necessary to ensure the partnership lasts the time defined by the partnerships. The final quality attribute was **acceptability**, which has been recognized as an essential attribute for collaboration, as for example in a surveillance system [20].

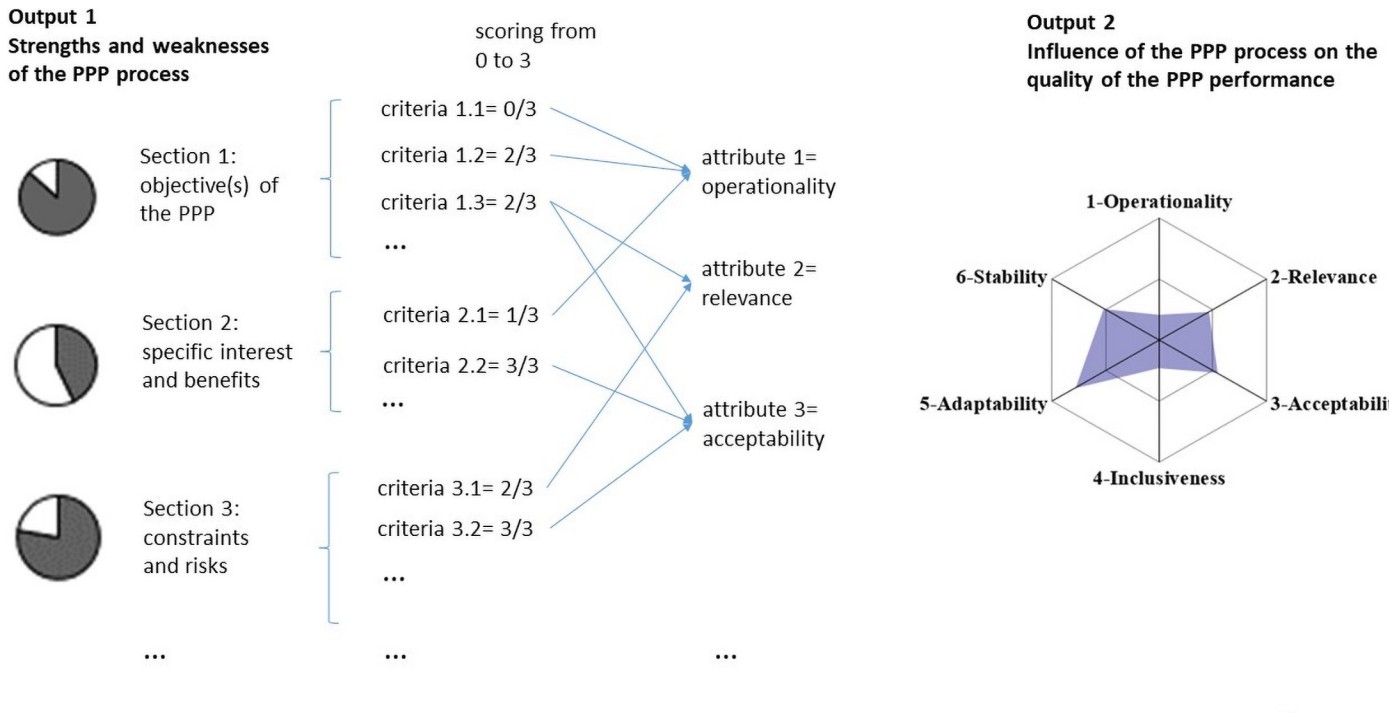

**Fig 2. Principle of the scoring process used in the tool, allowing two graphic outputs.** The graphic output 1 (the strengths and weaknesses of the structure of the process) represents the assessment of the ten sections using a set of evaluation criteria. The graphic output 2 (the influence of the process on the quality of the public-private partnership performance) represent the assessment of the six quality attributes, influenced by evaluation criteria. The scores of the evaluation criteria have been randomly assigned. PPP: public-private partnership.

Version 1 of the tool was improved thanks to the stakeholders' feedback from the PPP training workshops organized by OIE. Stakeholders pointed out that the evaluation criterion "achievement of goal(s) of the Veterinary Service" should be supplemented by another evaluation criterion on the goal(s) of the private service. They advised joint consideration of the funding and human resources, which constitute complementary inputs. Two evaluation criteria were then modified to "funding and human resource availability" and "funding and human resource allocation". They asked for clarification/simplification of some words, for example the term "externalities" which was revised to "cost to the society". They expressed the need for a self-assessment tool for implementation of the PPP field. The stakeholders perceived the tool as useful both to assess the quality of existing PPPs but also to assist them in planning new PPPs.

## Validation of the tool through the experts' elicitation process

In the first round of experts' elicitation, 45 out of the initial 48 evaluation criteria were validated. It was underlined that, even if relevant, the evaluation criteria may not be appropriate for all PPPs:

- "an early collaborative PPP in a country with little PPP uptake may be enabled by the absence of a degree of formality that would put off potential partners" (comment from a public expert during the 1st round of the experts' elicitation)

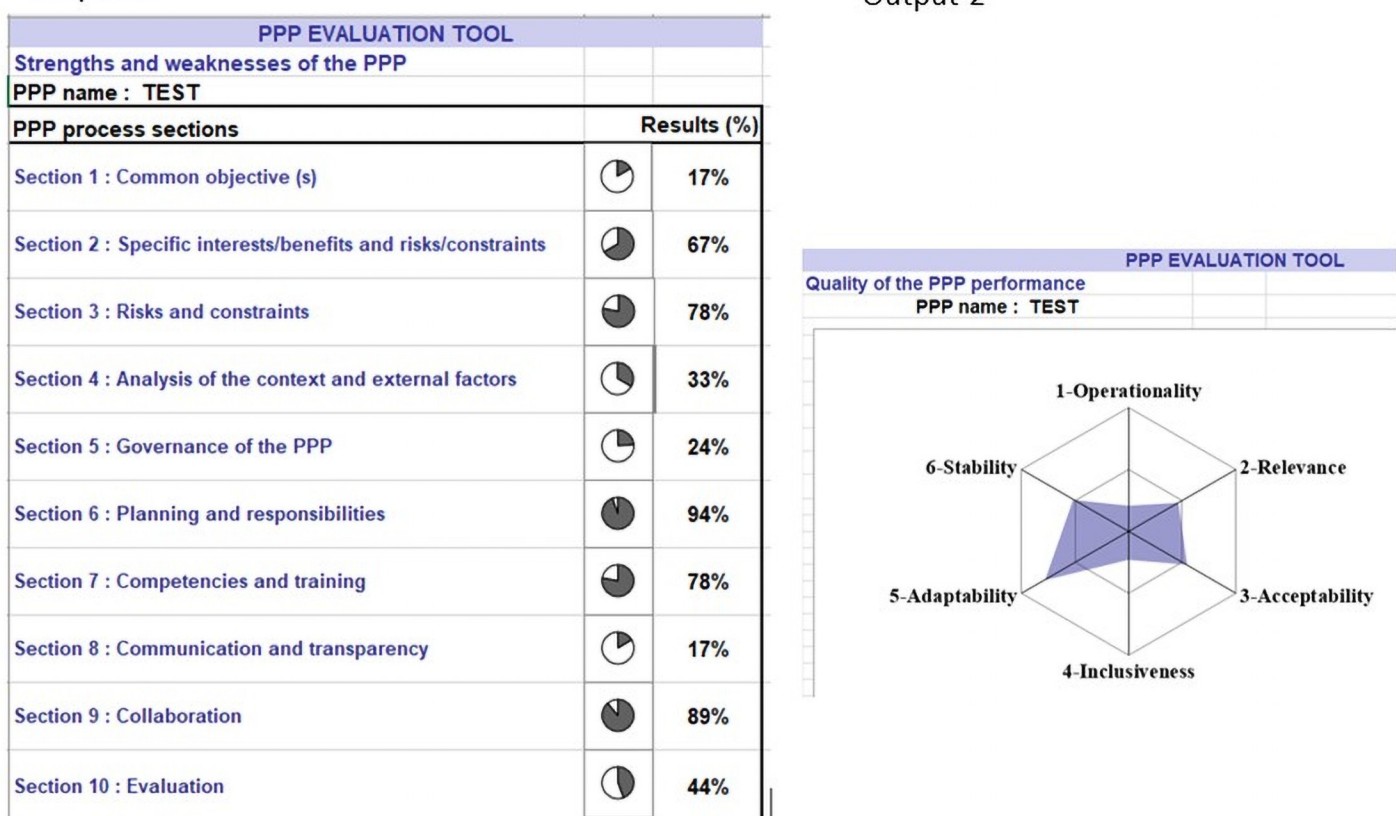

**Fig 3. The two graphic outputs of the evaluation tool for the public-private partnership process.** Graphic output 1 is a set of pie charts (the assessment of the sections), making it easy to identify the strengths and weaknesses of the process. Graphic output 2 is a spider chart (the assessment of the quality attributes), representing the influence of the process on the quality of the public-private partnership performance. The scores of the evaluation criteria have been randomly assigned.

Only 3 out of the 48 evaluation criteria were not considered as relevant by the experts: "shared decision making", "potential other partners" and "modalities of implementation of the PPP activities". Modifications and/or clarifications of those evaluation criteria were proposed based on the analysis of the experts comments and included in the second round. Seven evaluation criteria were merged with other evaluation criteria based on the expert's comments. Two new evaluation criteria were proposed and included in the second round ("joint work plan", "conflict of interest").

The six quality attributes were validated. Some of the levels of influence of evaluation criteria (which can influence more than one quality attribute) on the six quality attributes were validated (12/15 for operationality attribute, 6/8 for relevance attribute, 15/17 for acceptability attribute, 11/12 for inclusiveness, 7/8 for adaptability attribute and 14/16 for stability attribute). The levels of influence not validated were included in the second round. The experts also suggested adding some influence links between evaluation criteria and certain quality attributes; these proposals were also included in the second round (11 new influence links).

All the modifications and clarifications of the evaluation criteria (3/3) were validated in the second round. Three experts still mentioned that the evaluation criterion "shared decision making process" was not relevant:

- "how can we say that all decisions must be made in consultation with all PPP partners? Which level of decisions? Collaboration is time-consuming and costly and should be used when necessary, but not for all decisions" (comment from a catalyser expert during the 2nd round of the experts' elicitation).

The two new evaluation criteria were validated. Commenting on the question on the evaluation criterion "conflict of interest", one expert expressed concern that the tool may not pay sufficient attention to issues related to corruption, favoritism, unfair competition, consideration of the common good and the best interests of the population, as these risks could involve either private or public sector actors. A new evaluation criterion, "analysis of pre-existing PPP" was proposed during the second round and was included in the tool after validation by 4 members of the research team.

Almost all levels of influence of the evaluation criteria on the quality attributes were validated (15/19). The levels of influence that did not reach consensus were all between medium (score of 5) and high level (score of 10), therefore an arbitrary intermediate score was given to them (score of 7.5) (Fig 4).

Overall, 41 evaluation criteria were considered to highly influence at least one quality attribute; only 3 evaluation criteria influence the quality attributes with a medium or intermediary level only (3.3 "change of practices", 6.2 "potential other partners", 6.5 "modalities of implementation of the PPP activities") and none were considered not to influence the quality of the PPP performance at all (Table 3). The high level of influence of the evaluation criteria "change of practices" on the attribute "operationality" was selected by only 25% of the catalyser experts and 33% of the public partners, whereas it was selected by 50% of the private partners, and a medium level of influence was attributed.

## Application of the tool on a public-private partnership in Paraguay for the control of the foot-and-mouth disease

This PPP has existed since 2003 between the public Veterinary Services and a private foundation created by bovine producers. The private sector is a foundation recognized by a decree of the executive power and is responsible for coordinating and vaccinating the 15 million head of cattle. All these activities are supervised by the Veterinary Services. The PPP has evolved over the years, in terms of the partners involved and the type of governance. This PPP allowed Paraguay to obtain the status *FMD free with use of vaccination* from OIE. Paraguayan stakeholders, who have long experience of being involved in this PPP, found this tool comprehensive and the questions easy to understand. They acknowledged that, by implementing the tool, the group involved in the assessment process was able to address all the activities of the PPP.

It also raised important points, such as the future of this collaboration if vaccination stops (through the evaluation criteria 7.1 "Confidence in other partners' competencies and satisfaction of partners about their own competencies", 9.1 "Willingness to collaborate and partners' acceptance of their own roles", and 9.3 "Willingness for capacity building in PPPs"). Evaluation criterion 8.2 "agreement in resolution modalities in case of conflict between partners" had not been raised and the partners felt it was important to include it in their legal agreement. They revealed that the PPP represented a means to achieve their goal in a complex institutional environment (through evaluation criterion 4.2 "International, regional, national and local laws" and 4.3 "Potential threats of the PPP and mitigation").

The public partners of the Veterinary Services were afraid of losing influence by letting a private foundation take care of the vaccination campaign (this was captured in evaluation criterion 3.1 "Risks and constraints of getting involved in the PPP" and 5.7 "Adequacy with the Veterinary Services mandate"). Meanwhile, the private foundation feared its status might be

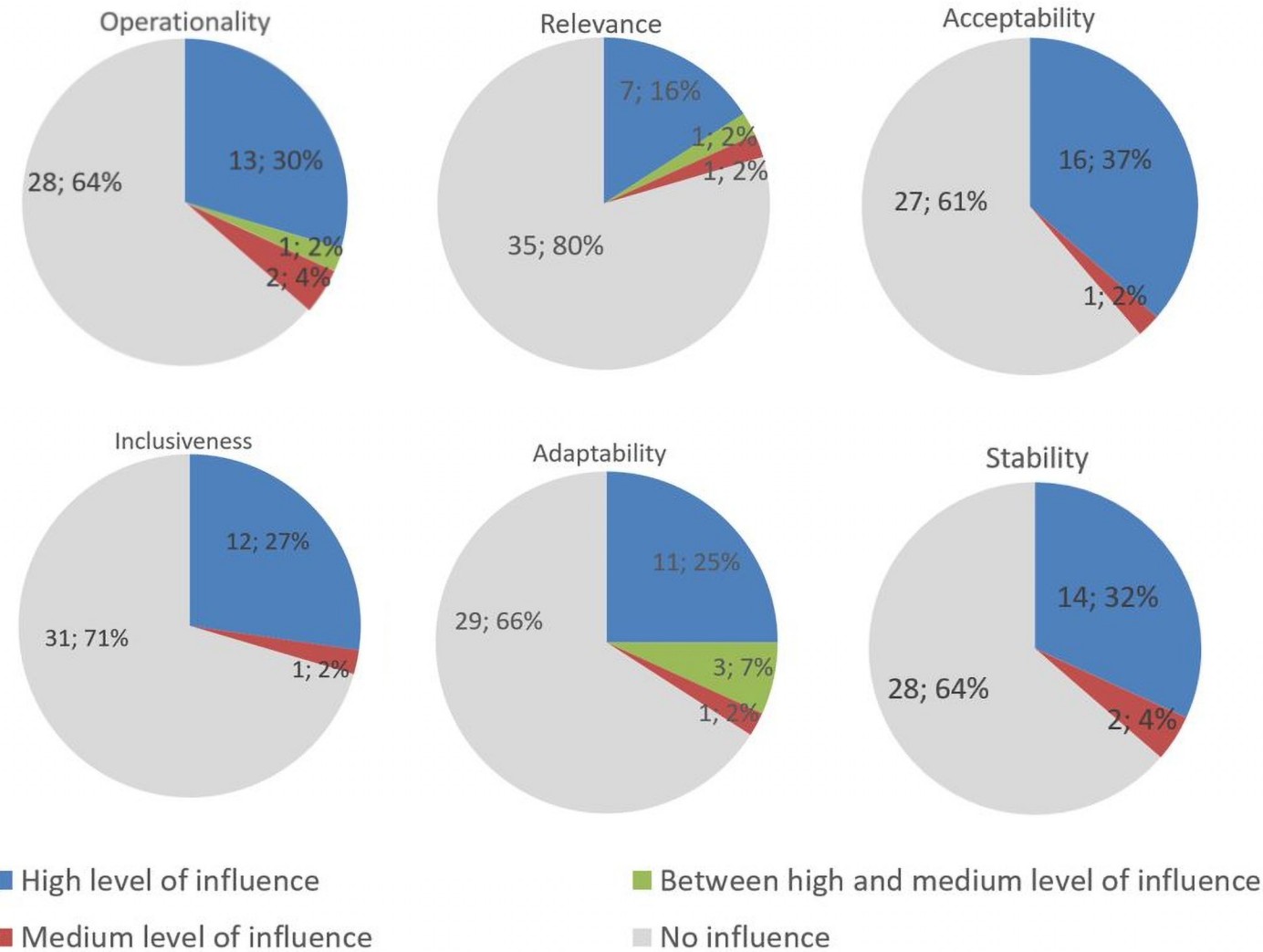

**Fig 4. Each of the six quality attributes are influenced by some evaluation criteria.** The level of influence of those evaluation criteria can be high (pie chart area in blue), between medium and high (pie chart area in green), medium (pie chart area in red). Some evaluation criteria do not influence the quality attribute (pie chart area in grey). The number and percentage of evaluation criteria per level of influence that influence each of the quality attributes are entered in the corresponding pie chart area.

erased in case of a change of political regime (evaluation criterion 3.1 "Risks and constraints of getting involved in the PPP"). Therefore, they reconsidered the status of the foundation, clarifying its roles and its range of action at the legislative level (this was captured in evaluation criterion 5.1 "Formalization of the PPP").

The PPP implemented for FDM vaccination enabled trainings for technicians at local level, resulting in an extension of the stakeholder network for the animal health value chain (captured in evaluation criterion 7.2 "Organisation of trainings and reinforcement of capacities" and 7.3 "Accessibility and frequency of trainings"). This network has, for example, led to reporting cases of bovine rabies in a village with rapid feedback of the information to Veterinary Services at the national level. The services provided by the PPP therefore exceed the initial objective of vaccination against FMD by reinforcing the Veterinary Services, and the tool was able to capture this element.

## Discussion

This study presents the development of a tool to evaluate the PPP process through the participation of relevant actors directly or indirectly involved in PPPs in the veterinary domain worldwide. To our knowledge this work is original and provides an assessment of the quality of the PPP process in the veterinary domain, addressing the question: "how, why and under which conditions does the PPP work?". This tool can help to evaluate and improve an ongoing PPP initiative but also to plan a new PPP under development. The tool can be used in an *ex ante* evaluation- during the PPP design phase, to help raise collective awareness of the challenges of PPP collaborations and to promote a more coordinated approach to collective actions [21]. The tool can also be used *in itinere*, when an initiative is already implemented, to promote partners' communication, good collaboration and to strengthen the PPP. The tool is freely accessible and placed under creative commons licence.

### Enabling dialogue between public-private partnership partners

The tool was developed in the same format as the Oasis tool, which demonstrated its ease of use, an important aspect to ensure its implementation in the field [22]. This format can also be compared to the Rubric tool, an easy-to-use tool for collaborative performance assessment [23]. The Rubric tool is constructed with the same two key components: a list of evaluation criteria and gradations of the quality of those evaluation criteria by people involved in the collaboration [23]. It was initially employed in educational sciences but has also demonstrated its effectiveness in other fields such as pest management [21]. This specific tool format facilitates the sharing of diverse perspectives and is adaptable to varied programs [21]. Like Rubric, the PPP evaluation tool developed here differs from a simple checklist, as each evaluation criterion requires gradations (from 0 to 3), involving discussion and precise justification of the expectations of the different stakeholders [24]. Asking the partners from the Paraguayan PPP to justify their choice of a score for each evaluation criterion indeed implied a process of dialogue between them, which facilitated reflection and analysis of the PPP. The use of the tool helped to clarify partner's expectations about various aspects of the PPP. This kind of tool allows stakeholders to make reliable judgements about their own work and identify room for improvement [25]. The scores given to each evaluation criterion are not as important as the dialogue between stakeholders during the evaluation. This tool can be seen as a means of mediation, helping to identify points of disagreement between partners, but also to clarify stakeholders' expectations and ways of improving. These are essential aspects in PPP best practices to ensure performances and impact of collaboration [2].

The tool can be used both for internal and external evaluation. A trained external evaluator expert can use this tool to evaluate any PPP process, but it is critical—as for any assessment—that the evaluation request arises from the stakeholders of the PPP themselves. The evaluator also needs to follow best evaluation practices, including objectivity and integration of multiple viewpoints [26]. This implies following a proper stakeholder mapping approach to ensure engagement with all the relevant stakeholders during the participatory interviews to capture diverse and representative viewpoints [27–29]. Mapping may include stakeholders who will use the evaluation results directly, who will support or maintain partnerships or who will be affected by the partnership's activities or assessment results [9]. Stakeholder mapping is therefore a pre-requisite step before implementing the tool. To ensure objectivity in the evaluation, the external evaluator would need to ensure the involvement of the stakeholders during the scoring process, rather than simply reflecting the prevailing expert view [23]. This tool can also be used during an internal evaluation process by the partners involved in the PPP for self-assessment of the quality of their PPP, also ensuring the involvement of all the relevant

stakeholders. This approach has the advantage of being inclusive; however, we argue that it would require either a previous training or a facilitation process for the partners by an evaluation expert to ensure proper use of the tool.

When using this tool, the evaluator should bear in mind that participatory approaches, including evaluation, cannot erase pre-existing social inequalities which may hamper the capacity of actors to express themselves freely. Genuine participation of all stakeholders may not be fully achieved, since power structures, inherent to social groups, will limit the free expression of marginalised people. Indeed these people may not be able to risk taking positions that run counter to those of power groups [30]. Trying to represent the diversity of viewpoints from stakeholders who influence, who are involved in or impacted by the PPP during the evaluation process is a real challenge. The use of this tool as well as participatory approaches can be a way to achieve this, but we argue that the limits of the evaluation process and results should be critically analyzed, emphasized, and expressed in a transparent manner by the evaluator. The risk of not doing so, would be to reinforce pre-existing power relations between stakeholders by only representing the dominant viewpoint [31].

## A generic tool to evaluate the quality of the process across different public-private partnership clusters

As mentioned before, three main clusters of PPP (transactional, collaborative and transformative) have been identified in the veterinary domain, depending on the type of private partner involved and the governance process [5]. However some PPPs are at the crossroad between clusters. The FMD control PPP in Paraguay, for example, is a mix between transactional PPP —private veterinarians and technicians are mandated and evaluated by the Veterinary Service to carry out the vaccination—and collaborative PPP—with the strong involvement of the producer association. Even though previous work has highlighted differences in obstacles depending on the PPP clusters, e.g.—the type of governance can represent an obstacle for collaborative and transformative PPPs, while the transactional PPP obstacles are mainly linked to lack of funding and human resources. Key success factors were not associated with any particular PPP type in the veterinary domain [5]. This indicates that the critical elements of the PPP process captured in this tool are similar across the clusters, which implies that PPP process evaluation could be generic across the different PPP types [7].

## The need for flexibility in public-private partnership evaluation

Each PPP in the veterinary domain, regardless of PPP cluster, needs to be adapted to the context; the evaluation process therefore needs to be flexible to ensure its relevance. This tool should not be used in a normative evaluation approach, and the evaluation criteria should not be seen as target objectives to be achieved.

For example, several evaluation criteria are linked to PPP formalization and naming the collaboration can increase the willing consent of partners [32] and support accountability [33]. However, several experts mentioned that too much formalization may hamper the development of the collaboration. Depending on the PPP to be evaluated, these evaluation criteria may not be relevant. Regarding the evaluation criteria related to the planning of PPP (section 6), planning can be done as a "deliberate approach", meaning that formal planning is carried out in advance, or as an "emergent approach", whereby precise planning emerges over time [14]. One approach is no better than the other. Another example is the evaluation criterion linked to law and regulation (evaluation criterion 4.2): institutional and political environment as well as other external factors are important for the PPP process and can strongly influence the initiative [14]; however, in accordance with the testimonies of Paraguayan stakeholders,

the external environment will not always determine collaborative action, and PPP may be a means to achieve objectives in a complex environment. Finally, an evaluation criterion related to inclusion of vulnerable groups in the planning process (6.3), and an evaluation criterion targeting shared decision making (5.3) were included in the tool. The protocol for PPP evaluation in Public Health also has a section targeting vulnerable groups, as a crucial aspect of World Health Organization programs is to enhance equity in health and well-being [34]. However, one expert mentioned that inclusion is not always the most appropriate way to take decisions and that shared decision making should be used when necessary. These examples demonstrate that flexibility in the evaluation process in adapting to the specific PPP context is essential to providing useful recommendations.

The tool presents a predefined list of evaluation criteria, allowing the users to review and challenge some aspects/elements of their collaboration process that they might not have considered *a priori*. For example, after mentioning the evaluation criterion "mechanism in place in case of conflict", the Paraguayan partners discussed the possibility of creating such a mechanism. Indeed, the aim of the tool is to be as complete as possible to cover the multiple types of PPP process which exists worldwide [5]. However, some evaluation criteria may not always be relevant in all situations and the tool allows for the use of 'not applicable' to remove evaluation criteria from the scoring process. This option further enhances the flexibility of the tool and limits its normative aspect.

It is also interesting to note that in the experts' elicitation, a smaller proportion of catalyzer and public experts, compared to private experts, considered that the evaluation criterion highly influence quality attribute operationality. This may be due to the fact that private actors in the veterinary domain (such as private veterinarians, producers) are those who are impacted by the change in practices in the field, whereas the catalyzers are actors operating in international organizations, and public actors, from the Veterinary Services in our sample, often operate at a central level. However, this result should be interpreted with caution in the case of public actors, as only three of them participated in the experts' elicitation. For some actors not operating in the field, it may be difficult to anticipate the difficulties encountered by actors in the field in implementing the modalities decided at central level. This underlines the importance of considering multiple points of view in our methodology for the development of the tool.

## The need to anticipate the risks of being involved in public-private partnerships

The OIE PPP handbook and the PPP reference guide from the World Bank both emphasize the need to compile a complete list of all risks associated with the project and to think about risk allocation [2, 35]. The different steps of this study (literature review, PPP regional training workshop and experts' elicitation) confirmed that partners need to clearly identify those risks in order to be able to limit them. The "negative cost to the society" (criterion 5.4) deals with the negative consequences of PPP, assuming that if the partners anticipate and undertake corrective action to prevent negative consequences of their partnership, the PPP will be more stable over time and its legitimacy in the eyes of society will be increased. Similarly, the Food and Agriculture Organisation guidelines to ensure good PPP practices within agricultural value chains proposes integrating the risks linked to the negative cost of a program (externalities) in the planning process to ensure sustainable value chains [36]. The risks of potential conflicts of interest were recurrently highlighted during this study (literature review, experts' elicitation). According to the World Bank, PPPs can represent a risk of corruption i.e. the misuse of public office for private gain [35]. Corruption seems to be favoured when privatizing certain state-owned enterprises [37]. Moreover, PPPs, like any contractual relationship, can be seen as a

"principal-agent" relationship in which the principal is the public partner (the public Veterinary Services) using the service of an agent, the private partner. This type of relationship involves differences of interest and asymmetrical information between the two contracting parties, with the practical impossibility for contracts to cover all possible cases and prevent all types of misconduct. Hence, partners having different interests are likely to develop opportunistic behaviour, taking advantage of asymmetries of information and loopholes in the contract [38]. Therefore, for some PPPs, the contract between the two parties, the legislative environment and the governance structure will require particular attention to limit such risks. In addition, the evaluation of the PPP process needs to take into account the institutional capacity of both public and private partner. Indeed, depending on the type of PPP in the veterinary domain, unequal power relations can be expected (representing a disadvantage for the public or the private sector) that will influence the governance process. For example, it is most important that both partners are able to clearly defend their own interests without any opportunistic behaviour while having the necessary degree of information symmetry during the negotiation phase [38]. When relevant and appropriate, PPPs should have a contract that is "clear, comprehensive" and that "creates certainty for the contracting parties" [35]. Given the complexity and uncertainty of the environment, the contract will also require flexibility to enable changing circumstances to be dealt with [35] and to provide modalities for the renegotiation of contracts [38].

Such issues are taken into account by the evaluation tool proposed in this study and its implementation can help identify weaknesses in the PPP process that would need to be deeply analyzed. For example, experts in legal frameworks from the OIE Veterinary Legislation Support Program can deeply analyse the legal framework and the Performance of Veterinary Services evaluation can identify the potential weaknesses of the institution and help to prevent risks [4, 39]. The tool helps to identify the strengths of the PPP process, as well as helping to promote partner engagement, transparency and trust, thereby limiting these risks. Regular PPP evaluations, e.g. using this tool, from the planning stage (*ex-ante* perspective), during the PPP (*in itinere*) until the end of the PPP (*ex post*), make it possible to promote good practices, improve the performance of PPPs and limit the potential risks associated.

## Conclusions

The PPP process evaluation tool developed in this study represents a necessary milestone for a more comprehensive evaluation of PPPs. The tool does not replace other types of evaluation such as context analysis, economic, or impact assessment. It enables, with limited financial means, stakeholder engagement bringing out discussions that help to identify the strengths and weaknesses of the PPP process. It is also intended that this tool will serve as a basis for developing targeted support on PPP in the veterinary domain in the context of the OIE PVS Pathway. Recommendations following the implementation of this tool may include the need for further evaluation or analysis by implementing other methods, such as deeper investigation of the legal framework, or the analysis of institutional capacities. An evaluation of the impacts of the PPP may also be pertinent to define relevant indicators to monitor the progress of the initiative and motivate the partners involved, to advocate for additional resources from investors, or to ensure trust. This can be done for example with impact pathway methodology, using the theory of change [40, 41].

PPP in the veterinary domain are widely implemented worldwide and are often complex, dynamic, multilevel systems [14]. This PPP process evaluation tool represents a straightforward approach to provide direction or positive changes by strengthening the partnership.

## Supporting information

**S1 File. Questionnaire of the first round of the experts' elicitation.**
(PDF)

**S2 File. Questionnaire of the second round of the experts' elicitation.**
(PDF)

**S3 File. The scoring guide of the evaluation tool for the public-private partnership process.**
(DOCX)

## Acknowledgments

The authors gratefully acknowledge the stakeholders from the PPPs in Ethiopia and in Paraguay, participants of the PPP regional training workshops organized by OIE, and the 27 experts who responded to the experts' elicitation.

## Author Contributions

**Conceptualization:** Mariline Poupaud, Nicolas Antoine-Moussiaux, Marisa Peyre.

**Data curation:** Mariline Poupaud.

**Formal analysis:** Mariline Poupaud.

**Investigation:** Mariline Poupaud.

**Methodology:** Mariline Poupaud, Marisa Peyre.

**Project administration:** Isabelle Dieuzy-Labaye, Marisa Peyre.

**Resources:** Isabelle Dieuzy-Labaye.

**Supervision:** Nicolas Antoine-Moussiaux, Marisa Peyre.

**Validation:** Nicolas Antoine-Moussiaux, Isabelle Dieuzy-Labaye, Marisa Peyre.

**Writing – original draft:** Mariline Poupaud.

**Writing – review & editing:** Nicolas Antoine-Moussiaux, Isabelle Dieuzy-Labaye, Marisa Peyre.

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
