## [Decision Letter · Decision Letter 0]

10 May 2021

An evaluation tool to strengthen the collaborative process of the public-private partnership in the veterinary domain

PONE-D-21-08118

Dear Dr. Poupaud,

We’re pleased to inform you that your manuscript has been judged scientifically suitable for publication and will be formally accepted for publication once it meets all outstanding technical requirements.

Kind regards,

Bilal Aslam, PhD

Academic Editor

PLOS ONE

Reviewers' comments:

Reviewer's Responses to Questions

**Comments to the Author**

1. Is the manuscript technically sound, and do the data support the conclusions?

Reviewer #1: Yes

Reviewer #2: Yes

2. Has the statistical analysis been performed appropriately and rigorously? 

Reviewer #1: No

Reviewer #2: Yes

3. Have the authors made all data underlying the findings in their manuscript fully available?

Reviewer #1: Yes

Reviewer #2: Yes

4. Is the manuscript presented in an intelligible fashion and written in standard English?

Reviewer #1: Yes

Reviewer #2: Yes

5. Review Comments to the Author

Reviewer #1: A very well designed study that outlines the objectives, methods, results and conclusions very effectively. I am confident this study will help and lay out guidance principles for a much needed Public-private partnership in Veterinary field. This study has much larger applications in a global setting.

Reviewer #2: The Manuscript is well written in standard English and intelligible fashion. The authors have grip to the topic and conducted the study in scientific way. I acknowledge the authors for their contribution towards the Veterinary Field

6. PLOS authors have the option to publish the peer review history of their article (what does this mean?). If published, this will include your full peer review and any attached files.

Reviewer #1: No

Reviewer #2: No

---

## [Editor Report · Acceptance letter]

20 May 2021

PONE-D-21-08118 

An evaluation tool to strengthen the collaborative process of the public-private partnership in the veterinary domain 

Dear Dr. Poupaud:

I'm pleased to inform you that your manuscript has been deemed suitable for publication in PLOS ONE. Congratulations! Your manuscript is now with our production department. 

Kind regards, 

on behalf of

Dr. Bilal Aslam 

Academic Editor

PLOS ONE